# The Homogeneous Gas-Phase Formation Mechanism of PCNs from Cross-Condensation of Phenoxy Radical with 2-CPR and 3-CPR: A Theoretical Mechanistic and Kinetic Study

**DOI:** 10.3390/ijms23115866

**Published:** 2022-05-24

**Authors:** Zhuochao Teng, Yanan Han, Shuming He, Mohammad Hassan Hadizadeh, Qi Zhang, Xurong Bai, Xiaotong Wang, Yanhui Sun, Fei Xu

**Affiliations:** 1Environment Research Institute, Shandong University, Qingdao 266237, China; zct@mail.sdu.edu.cn (Z.T.); hyn@mail.sdu.edu.cn (Y.H.); hadizadeh.mh@sdu.edu.cn (M.H.H.); sophia@mail.sdu.edu.cn (Q.Z.); 202033064@mail.sdu.edu.cn (X.B.); wangxiaotongjy@163.com (X.W.); 2School of Environment, Hangzhou Institute for Advanced Study, University of Chinese Academy of Sciences, Hangzhou 310024, China; heshuming20@mails.ucas.ac.cn; 3College of Environment and Safety Engineering, Qingdao University of Science & Technology, Qingdao 266042, China; sunyh0532@126.com

**Keywords:** polychlorinated naphthalenes, chlorophenols, formation mechanisms, rate constants, DFT calculations

## Abstract

Chlorophenols (CPs) and phenol are abundant in thermal and combustion procedures, such as stack gas production, industrial incinerators, metal reclamation, etc., which are key precursors for the formation of polychlorinated naphthalenes (PCNs). CPs and phenol can react with H or OH radicals to form chlorophenoxy radicals (CPRs) and phenoxy radical (PhR). The self-condensation of CPRs or cross-condensation of PhR with CPRs is the initial and most important step for PCN formation. In this work, detailed thermodynamic and kinetic calculations were carried out to investigate the PCN formation mechanisms from PhR with 2-CPR/3-CPR. Several energetically advantageous formation pathways were obtained. The rate constants of key elementary steps were calculated over 600~1200 K using the canonical variational transition-state theory (CVT) with the small curvature tunneling (SCT) contribution method. The mechanisms were compared with the experimental observations and our previous works on the PCN formation from the self-condensation of 2-CPRs/3-CPRs. This study shows that naphthalene and 1-monochlorinated naphthalene (1-MCN) are the main PCN products from the cross-condensation of PhR with 2-CPR, and naphthalene and 2-monochlorinated naphthalene (2-MCN) are the main PCN products from the cross-condensation of PhR with 3-CPR. Pathways terminated with Cl elimination are preferred over those terminated with H elimination. PCN formation from the cross-condensation of PhR with 3-CPR can occur much easier than that from the cross-condensation of PhR with 2-CPR. This study, along with the study of PCN formation from the self-condensation 2-CPRs/3-CPRs, can provide reasonable explanations for the experimental observations that the formation potential of naphthalene is larger than that of 1-MCN using 2-CP as a precursor, and an almost equal yield of 1-MCN and 2-MCN can be produced with 3-CP as a precursor.

## 1. Introduction

Polychlorinated naphthalenes (PCNs) are a class of mainly industrial chemicals de-rived from fusing diaromatic hydrocarbons, which include 75 congeners containing one to eight chlorine atoms per naphthalene molecule. PCNs are often called dioxin-like compounds, and they show similar physical properties, chemical properties, and toxicity to polychlorinated dibenzo-p-dioxins and dibenzofurans (PCDD/Fs), as well as polychlorinated biphenyls (PCBs) [1,2,3]. PCNs are ubiquitous in the environment and have been detected in the polar environment due to their potential for persistence, lipophilicity, bioaccumulation, and long-distance atmospheric migration [4,5,6]. Environmental PCNs could contain three categories of sources, including PCN technical products, PCNs as trace impurities in PCBs technical mixtures, and unintentionally produced PCNs from industrial thermal and combustion processes [2,7,8]. The production and use of PCNs have been discontinued in most countries since the 1980s [9]. Currently, the unintentional formation of PCNs as byproducts from thermal and combustion processes, such as waste incineration, secondary copper smelting, iron-steel production, chloralkali industries, steel production, nonferrous smelting, and coking industries [10,11,12,13,14,15,16,17], was reported as the current primary source of PCNs, accounting for 80% of PCN emissions [1]. The average concentration of PCNs in stack gas samples from a municipal solid waste incinerator was reported to be 710 ng/m^3^ [18]. The environmental pollution and human health hazards caused by PCNs have raised considerable public concern around the world in recent years. Thus, PCNs were listed in annexes A and C of the Stockholm Convention on Persistent Organic Pollutants (POPs) in 2015, which means that the intentional production of PCNs is forbidden, and the inadvertent release of PCNs during various industrial processes should be reduced.

Chlorophenols (CPs) are a major group of chlorinated chemical pollutants of environmental concern that are widely used in the chlorinated bleaching of pulp, oil refining, fungicides, pharmaceuticals, herbicides, and dyes. CPs exhibit high toxicity and are able to produce mutagenic, carcinogenic, estrogenic, and low biodegradable effects in human beings, and are even unsafe for the environment and ecological system as well [19,20,21,22,23,24,25]. High concentrations of CPs (0.6~141.0 μg/Nm^3^) were detected in the stack gas emissions during the MSW incineration process [26]. Due to the different substitution patterns of phenol (Ph), CPs have 19 isomers. Among them, 2-chlorophenol (2-CP) and 3-chlorophenol (3-CP) are the most typical monochlorophenols. In combustion and thermal processes, chlorinated phenoxy radicals (CPRs) can be produced through the loss of phenoxyl-hydrogen via the unimolecular cleavage of the O-H bond or abstracted by the active radicals H, OH, O(^3^P), and Cl. CPs have been demonstrated to be predominant precursors in the formation of PCNs [27,28,29,30,31,32]. Radical/radical dimerization of CPRs is the initial and crucial pathway of homogeneous gas-phase formation of PCNs. Besides CPs, phenol is typically much more abundant than CPs in municipal waste incinerators. The concentration of phenol in the flue gases of municipal waste incineration was measured to be 30–100 times higher than that of total CPs [33]. At high temperatures, phenol can be obtained from CPs by losing Cl atoms or from benzene by adding OH groups. The cross-condensation of phenoxy radical (PhR) with CPRs is also responsible for the distribution of PCN isomers, especially for the low chlorinated PCN homologues.

Several exhaustive PCN formation mechanisms using experimental and theoretical methods have been proposed [27,28,29,30,31,32]. Kim et al. performed a laminar flow reactor experiment at 600 °C combustion temperature to study the PCN formation from 2-CP and 3-CP as precursors. They found that PCNs were produced via C–C connection on unchlorinated *ortho* CPR positions, forming a chlorinated o,o′-dihydroxybiphenyl (chloro-DOHB) intermediate [28,29,30]. For one thing, chloro-DOHB may form PCDFs via H migration, isomerization and H_2_O leave [28,29,30]. For another, chloro-DOHB can form chlorinated dihydrofulvalene by two carbon monoxide elimination steps and two ring close reactions, followed by PCN formation [28,29,30]. This mechanism can explain the more similar formation connection of PCN with PCDF rather than PCDD in thermal processes. Based on radical/radical gas-phase routes, a series of experimental and theoretical studies of PCN formation mechanisms from the CPs as precursors have been carried out [27,28,29,30,31,32], and the theoretical results agree well with the experimental observations. For example, Kim et al. performed a laminar flow reactor experiment at 600 °C combustion temperature to study the PCN formation from 2-CP and 3-CP as precursors and found that the yield of monochlorinated naphthalenes (MCNs) is larger than that of dichlorinated naphthalenes (DCNs) [28,29]. This has been verified by the theoretical study of PCN formation from self-condensations of 2-CPRs/3-CPRs, in which pathways that ended with the elimination of Cl were found to be dominant over pathways that ended with the elimination of H for PCN formation [31,32]. Moreover, in Kim et al.’s experiments, they found that PCNs can form along with PCDFs, but the yield of total PCNs is much lower than that of PCDFs [28,29]. Theoretical studies provided reasonable explanations, including that PCN formation has more elementary steps than PCDF formation, and the rate-determining step for PCN formation has a higher potential barrier than that for PCDF formation [31,32]. In addition, both the experimental and theoretical studies demonstrated that the chlorine substitution pattern has a significant effect on the isomer patterns and the formation potential of PCNs [28,29,30,31,32]. Despite the fact that many experimental observations can be verified by the theoretical studies, several experimental and theoretical results have not reached a consensus. For example, in Kim et al.’s experiment relating to PCN formation from 2-CP as a precursor, it has been found that the yield of naphthalene is higher than that of 1-MCN [28,29]. However, our previous theoretical study of PCN formation from the self-condensation of 2-CPRs showed that the formation potential of naphthalene is much lower than that of 1-MCN [31]. In the formation of PCN from 3-CP as a precursor, Kim et al. observed almost equal yields for 1-MCN and 2-MCN, while we have previously shown that the 1-MCN formation has higher formation potential than that of 2-MCN from the self-condensation of 3-CPRs [32]. It should be noted that no formation route of naphthalene from the self-condensation of 3-CPRs was found in our previous theoretical study [32], which was also largely observed in Kim et al.’s experiment using 3-CP as a precursor [28,29]. Thus, considering the high phenol concentration detected in Kim et al.’s experiment [28,29] and the formation possibility of naphthalene from phenol, the formation mechanisms of PCNs from the cross-reactions of phenol with CPs under high-temperature conditions are significant.

Due to the high toxicity of PCNs and the lack of an effective experimental detection method for some short-life intermediates, the mechanism of PCNs is still not clear. Quantum chemical calculation can be used to research highly toxic compounds, predict the feasibility of a reaction route, and confirm the priority of the products. In addition, The PCN formations from the self-condensation of 2-CPRs and 3CPRs are accompanied by the cross-condensation of PhR with 2-CPR, and PhR with 3-CPR, resulting in the products being mixed and difficult to distinguish in the experiment [28,29]. Hence, in this work, we carried out a comprehensive quantum calculation on the PCN gas-phase formation from the cross-condensation of PhR with 2-CPR, and PhR with 3-CPR in high-temperature pyrolysis and combustion conditions. Numerous energetically advantageous formation routes were proposed. Moreover, the reaction priority of different PCN formation pathways and isomer patterns of PCN products from cross-condensation of PhR with 2-CPR/3-CPR were discussed and compared with the experimental observations. Furthermore, rate constants for the key elementary reactions over 600~1200 K were evaluated. The final goal was to compare the formation potential of PCN products from the self-condensation mechanism of 2-CPRs/3-CTPRs and the cross-condensation mechanism of phenol with 2-CPRs/3-CPRs.

## 2. Results

### 2.1. Formation of PhR, 2-CPR, and 3-CPR from Phenol, 2-CTP, and 3-CP Molecules

The formation of phenoxy radical (PhR), 2-chlorophenoxy radical (2-CPR), and 3-chlorophenoxy radical (3-CPR) derived from phenol (Ph), 2-chlorophenol (2-CP), and 3-chlorophenol (3-CP) is the initial and key step in the formation of naphthalene and monochlorinated naphthalenes (MCNs). Dimerization of CPRs or cross-condensation of PhR with CPR can form PCNs. In combustion and thermal processes, these radicals may be generated by means of H extraction reactions of CPs by the active H, OH, Cl, or O(^3^P) radicals, which exist abundantly in high-temperature conditions. The potential barriers (Δ*E*) and the reaction heats (Δ*H*), which are calculated at the MPWB1K/6-311 + G(3df,2p)//MPWB1K/6-31 + G(d,p) level for the formation of PhR, 2-CPR, and 3-CPR from Ph, 2-CP, and 3-CP abstracted by the H, OH, Cl, or O(^3^P) are given in Appendix A. All the abstraction steps are strongly exothermic. In particular, the potential barriers of CPs and phenol with OH/Cl radicals are negative due to the ZPE (zero-point energy) correction and the existence of pre-reactive complexes. The reaction potential energy profile of phenol and OH radical is given in Appendix A. The CP and OH radical will form the pre-reaction complex (CP-OH) first; a transition state can then be produced. The transition state has a negative potential barrier relative to the reactants, but a positive potential barrier relative to the pre-reaction complex [34].

### 2.2. Formation of Chloro-Bicyclopentadienyl from Cross-Condensation of PhR with 2-CPR/3-CPR

PCN formation from the cross-condensation of PhR with 2-CPR/3-CPR contains two processes. The first process is the formation of chloro-bicyclopentadienyl (IM5 and IM13) from the cross-condensation of PhR with 2-CPR (Figure 1) or the formation of chloro-bicyclopentadienyl (IM5 and IM28) from the cross-condensation of PhR with 3-CPR (Figure 2). The second process is PCN formation from subsequent reactions of chloro-bicyclopentadienyls (direct abstraction pathways in Figure 2, Figure 3, Figure 4 and Figure 5 and then first shift abstraction pathways in Figure 6). The configurations of the transition states involved in one typical route of PCN formation are depicted in Figure 7. Imaginary frequencies, zero-point energies, and total energies for all the transition states involved in the formation of PCNs from the cross-condensation of PhR with 2-CPR/3-CPR are shown in Appendix A. The cartesian coordinates for the intermediates and transition states involved in PCN formation from the cross-condensation of PhR with 2-CPR/3-CPR are listed in Appendix A.

Figure 1 and Figure 2 demonstrate chloro-bicyclopentadienyl formation routes embedded with the potential barriers Δ*E* (kcal/moL) and reaction heats Δ*H* (kcal/moL) from the cross-condensation of PhR with 2-CPR and the cross-condensation of PhR with 3-CPR, respectively, e.g., the first process of PCN formation. All pathways include five elementary steps: (1) C-C coupling; (2) first phenolic ring opening; (3) first CO elimination; (4) second phenolic ring opening; (5) second CO elimination. The first/second CO elimination step is a synergetic reaction, e.g., CO loss and five-member ring formation occur at the same time. The C-C coupling step is barrierless and strongly exothermic, which is *ortho*’*ortho* coupling of CPRs to form an o,o′-dihydroxybiphenyl (DOHB) intermediate. As shown in Figure 1, three chloro-bicyclopentadienyls (IM5 and IM13) from four possible pathways (pathways 1–4) are proposed from the cross-condensation of PhR with 2-CPR. In Figure 1, in the first step of pathways 1 and 2, the coupling of two carbon (hydrogen)-centered radical mesomers (CH/CH) forms DOHB intermediate IM1, which finally produces IM5 with two *ortho* carbons connected by H, as shown in Figure 1. In the first step of pathways 3 and 4, the recombination of the carbon (hydrogen)-centered radical mesomer with the carbon (chlorine)-centered radical mesomer (CH/CCl) forms DOHB intermediate IM9, resulting in IM13 with one *ortho* carbon connected by H and another with Cl. In Figure 2, three chloro-bicyclopentadienyls (IM5 and IM28) from four possible pathways (pathways 5–8) are proposed from the cross-condensation of PhR with 3-CPR. All four pathways start with CH/CH coupling. Pathways 5 and 6 occur via the same DOHB intermediate IM17 and form the same chloro-bicyclopentadienyl intermediate IM5. Pathways 7 and 8 occur via the same DOHB intermediate IM24 and form the same chloro-bicyclopentadienyl intermediate IM28.

### 2.3. Formation of PCNs from Pursuant Reactions of Chloro-Bicyclopentadienyls

#### 2.3.1. Formation of PCNs from Pursuant Reactions of Chloro-Bicyclopentadienyls by Direct H/Cl Abstraction Routes

Figure 3, Figure 4 and Figure 5 show the PCN formation routes from subsequent reactions of IM5, IM13, and IM28 embedded with the potential barriers Δ*E* (kcal/moL) and reaction heats Δ*H* (kcal/moL). All the pathways start with the direct H/Cl abstraction of chloro-bicyclopentadienyls by H and OH radicals. In Figure 3, three PCN isomers (naphthalene, 1-MCN, and 2-MCN) are obtained via six probable PCN formation pathways (pathways 9–14) from pursuant reactions of IM5. In Figure 4, two PCN isomers (naphthalene and 1-MCN) are obtained via three probable PCN formation pathways (pathways 15–17) from pursuant reactions of IM13. In Figure 5, two PCN isomers (1-MCN and 2-MCN) are obtained via five probable PCN formation pathways (pathways 18–22) from pursuant reactions of IM28. The potential barriers to the H abstraction step of chloro-bicyclopentadienyls by OH are negative due to the ZPE (zero-point energy) correction and the existence of pre-reactive complexes. In Figure 3, Figure 4 and Figure 5, the potential barriers of the H abstraction of chloro-bicyclopentadienyls by the H radical are larger than those created by the OH radical, which indicates that the H abstraction of chloro-bicyclopentadienyls by the OH radical is more preferred than those by the H radical. On the contrary, the Cl abstraction of chloro-bicyclopentadienyls by the OH radical is less favored than those by the H radical. It is necessary to compare the H and Cl abstraction of chloro-bicyclopentadienyls by the H/OH radicals. In Figure 3, Figure 4 and Figure 5, the H abstraction by the H/OH radicals of chloro-bicyclopentadienyls have lower potential barriers than the Cl abstraction by the H/OH radicals. Thus, the H abstraction of chloro-bicyclopentadienyls is more liable to occur than the Cl abstraction.

#### 2.3.2. Formation of PCNs from Pursuant Reactions of Chloro-Bicyclopentadienyls by First H Shift and Then Abstraction Routes

Besides the directly abstracted pathways of IM5 and IM28 shown in Figure 3, Figure 4 and Figure 5, these chloro-bicyclopentadienyls can also be caused by the intramolecular rearrangement of 1,5-sigmatropic H shifts to the *ortho*-carbon of the cyclopentadiene ring before the *ortho*-carbon H atom is abstracted by the H, OH, and Cl radicals. This mechanism was inferred by Kim at al. according to his experimental observations [28,29,30]. Here, the H shift steps of IM5 and IM28 were studied using the quantum chemistry method, as presented in Figure 6. As shown in Figure 6, the H shift can occur in IM5 via one mode with the potential barrier of 26.12 kcal/moL and a reaction heat of 1.22 kcal/moL, while IM28 can occur via the three-mode, with a potential barrier of 21.76~23.12 kcal/moL and a reaction heat of −3.26~−2.64 kcal/moL. The H shift from IM5 has a higher potential barrier and is less exothermic than those from IM28, which means that the H shift from IM5 can occur more easily than those from IM28. The following H abstraction step from IM65 initiated by H and OH have lower potential barriers than those from IM66-IM68, initiated by H and OH. For example, the potential barrier of the H abstraction step of IM65 abstracted by H is 4.13 kcal/moL, whereas the potential barrier of the H abstraction (by H) step of IM66, IM67, and IM68 are 6.84, 7.34, and 6.01 kcal/moL, respectively. This indicates that the H abstraction step from IM65 by H and OH are more energetically favored than those from IM66, IM67, and IM68 by H and OH. However, the H abstraction step from IM65 by Cl is less favored than those from IM66, IM67, and IM68.

### 2.4. Rate Constant Calculations

It is difficult to experimentally determine the rate constants for the elementary reactions that lead to PCN formation, especially the reactions involving the short-life radical intermediates, because of the high toxicity of PCNs, the lack of appropriate experimental conditions, and the difficulty of detecting PCNs. In such a situation, an alternative method can be used to calculate the rate constant or other dynamic information directly based on the quantum calculations of electronic structure, energy data, matrices of force constants, Hessian matrices, and coordinates of each stationary and nonstationary points. In this study, the canonical variational transition state theory (CVT) and small-curvature tunneling (SCT) method [35,36,37,38] were used to calculate the rate constants for the main elementary steps involved in the PCN formation pathways from the cross-condensation of PhR with 2-CPR/3-CPR over 600~1200 K. The range covers the possible formation temperature of PCN formation in combustion and thermal processes. Actually, the CVT/SCT method has been successfully used by members of our group in previous studies involved in the formation of dioxin-like compounds, as well as PCNs [31,32], and it is verified to be an efficient method to calculate the rate constants after accuracy testing in these studies.

The calculated CVT/SCT rate constant values at a given temperature are listed in Appendix A, for chloro-bicyclopentadienyl formation routes from the cross-condensation of PhR with 2-CPR/3-CPR, and in Appendix A, for PCN formation routes from chloro-bicyclopentadienyls. To obtain the necessary factors, pre-exponential factors, and activation energy, the rate constants-temperature relationships are fitted into Arrhenius formulas, which are given in Table 1 and Table 2. The obtained kinetic data can be put into the PCN controlling and formation models as important parameters, which have been widely used to control the potential outcomes of PCNs regarding the environment and to predict the PCN formation production in combustion and thermal processes.

## 3. Discussion

### 3.1. Formation of PhR, 2-CPR, and 3-CPR from Phenol, 2-CTP, and 3-CP Molecules

In Appendix A, the phenoxyl-hydrogen abstractions from phenol by the H, OH, Cl, or O(^3^P) have lower potential barriers and are more exothermic than those from 2-CP and 3-CP, respectively, e.g., the reactivity of the O-H bond in phenol is stronger than that in 2-CP and 3-CP. For example, the potential barriers of phenoxyl-hydrogen abstractions from phenol by H is 11.73 kcal/moL, which is lower than those of 13.80 and 12.51 kcal/moL from 2-CP and 3-CP by H, respectively. In addition, the reaction heat of phenoxyl-hydrogen abstractions from Ph by H is −13.98 kcal/moL, which is more exothermic than those of 12.01 and 12.94 kcal/moL from 2-CP and 3-CP by H, respectively. Since the influence of the chlorine substitutions in polychlorinated phenols can roughly be described as additive effects of the *ortho* and *meta*. Chlorine in an aromatic ring is traditionally recognized as an electron-withdrawing group. The inductive effect of the electron-withdrawing chlorine and the intramolecular hydrogen bonding may ultimately be responsible for the reactivity of the O-H bonds in chlorophenols. In addition, the potential barriers of the phenoxyl-hydrogen abstraction from 2-CP by the H, OH, Cl, or O(^3^P) are higher than those from 3-CP, which means that phenoxyl-hydrogen abstraction from 3-CP is easier to obtain than that from 2-CP. Finally, the reactivity of the O-H bonds in Ph, 2-CP, and, 3-CP occurs in the order of 2-CP < 3-CP < Ph.

### 3.2. Formation of Chloro-Bicyclopentadienyl from Cross-Condensation of PhR with 2-CPR/3-CPR

In Figure 1, the ranking for the exothermic values of the two C-C coupling modes is as follows: CH/CH (−10.74 kcal/moL in pathways 1 and 2) > CH/CCl (−3.25 kcal/moL in pathways 3 and 4). This can be explained by the steric effect that the Cl atom is larger than the H atom. In addition, the largest barriers in pathways 1–4 are 48.18 kcal/moL, 51.53 kcal/moL, 52.90 kcal/moL, and 54.56 kcal/moL, respectively. Thus, considering the two aspects, the ranking for the chloro-bicyclopentadienyl formation potential from the cross-condensation of PhR with 2-CPR is as follows: pathway 1 > pathway 2 > pathway 3 > pathway 4. Pathway 1 is most favorable, resulting in the formation of IM5. The formation potential of IM13 is less than that of IM5. If both *ortho*-positions of the phenol are substituted with chlorine atom, the formation of PCNs is almost completely inhibited.

In Figure 2, no CH/CCl coupling mode exists, and the ranking for the exothermic values of the two CH/CH coupling modes is as follows: CH/CH (−13.05 kcal/moL in pathways 5 and 6) < CH/CH (−13.61 kcal/moL in pathways 7 and 8). In addition, the largest barriers in pathways 5–8 are 49.10 kcal/moL, 50.11 kcal/moL, 46.79 kcal/moL, and 49.11 kcal/moL, respectively. Thus, considering the two aspects, the ranking for the chloro-bicyclopentadienyl formation potential from the cross-condensation of PhR with 3-CPR is as follows: pathway 7 > pathway 8 > pathway 5 > pathway 6. As a result, pathway 7 is the most favorable, resulting in the formation of IM28. Meanwhile, it should be noted that the formation potential of IM5 is less than that of IM28.

### 3.3. Formation of PCNs from Pursuant Reactions of Chloro-Bicyclopentadienyls

#### 3.3.1. Formation of PCNs from Pursuant Reactions of Chloro-Bicyclopentadienyls by Direct H/Cl Abstraction Routes

In Figure 3, the first three elementary steps in pathways 9 and 10 are exactly the same. The differences between pathway 9 and pathway 10 occur in the last 2 or 3 steps. Pathway 9 is ended with the H elimination step. The second 6-member ring formation and H elimination occur separately, with the potential barriers of 15.37 and 15.59 kcal/moL and reaction heat of −16.92 and 3.16 kcal/moL, respectively. Different from pathway 9, the second 6-member ring formation and Cl elimination in pathway 10 occur synergistically. Thus, pathway 10 has one step less than pathway 9. In addition, the Cl elimination in pathway 10 has lower potential barriers (12.42 kcal/moL) and is more exothermic (−27.67 kcal/moL) than the second 6-member ring formation and H elimination in pathway 9. Thus, Cl elimination in pathway 10 can occur more easily than the second 6-member ring formation and H elimination steps in pathway 9. The product naphthalene of pathway 10 has more formation potential than that of the product 1-MCN of pathway 9. Pathways 11 and 12 have the same first three steps, and are all ended with the H elimination steps. The H elimination in pathway 11 (Δ*E* 16.08 and Δ*H* 3.72) has a lower potential and is less endothermic than that in pathway 12 (Δ*E* 16.22 and Δ*H* 3.82). Thus, pathway 11 is favored over pathway 12. For the same reason, pathway 13 is favored over pathway 14. Comparing the reaction potentials of pathways 10, 11, and 13 provides some interesting insights into how these pathways work. The rate-determining steps of these three pathways are the first 6-member ring formation steps, with the potential barriers of 23.22, 27.34, and 21.95 in pathways 10, 11, and 13, respectively. Comparing pathways 11 and 13, the rate-determining step of pathway 11 can occur by crossing higher potential barriers than those of pathway 13. In addition, the H elimination step in pathway 11 (Δ*E* 16.08 and Δ*H* 3.72) has a higher potential barrier and is more endothermic than that in pathway 13 (Δ*E* 15.59 and Δ*H* 3.16). Thus, pathway 13 is energetically preferred to pathway 11, i.e., 1-MCN is more abundantly produced than 2-MCN. Comparing pathways 10 and 13, the rate-determining step of pathway 10 has higher potential barriers than that of pathway 13. However, pathway 10 has one step less than pathway 13, and the Cl elimination step in pathway 10 has a lower potential barrier and is more exothermic than the H elimination step in pathway 13. Thus, pathway 10 and pathway 13 are competitive, i.e., both naphthalene and 1-MCN are the main products of the pursuant reactions of IM5.

For the same reason, in Figure 4, the rate-determining steps of pathways 15 and 16 have lower potential barriers than that of pathway 17. In addition, pathway 16 has the least elementary steps. Thus, pathway 16 is the most favored route for the secondary reactions of IM13, and naphthalene is the main product of the pursuant reactions of IM13. In summary, the main PCN products from the cross-condensation of PhR with 2-CPR are naphthalene and 1-MCN. This agrees well with Kim et al.’s experimental observation that naphthalene and 1-MCN is the main PCN product from 2-CP as a precursor, and the formation potential of 1-MCN is higher than that of 2-MCN [28,29]. In Kim et al.’s experiment, more naphthalene can be observed than 1-MCN using 2-CP as precursor [28,29]. However, our previous theoretical study of PCN formation from the self-condensation of 2-CPRs showed that the formation potential of naphthalene is much lower than that of 1-MCN [31]. For example, for the naphthalene formation from the self-condensation of 2-CPRs, the potential barrier for the rate-determining (first 6-member ring formation) step and the Cl elimination step are 23.52 and 12.11 kcal/moL, respectively [32]. For the 1-MCN formation from the self-condensation of 2-CPRs, the potential barrier for the rate-determining (first 6-member ring formation) step and the Cl elimination step are 22.34 and 10.85 kcal/moL, respectively [31]. This study provides an additional naphthalene formation mechanism, which can be obtained from the cross-condensation of PhR and 2-CPR. In addition, the self-condensation of PhRs may yield a contribution to the naphthalene formation based on the high concentration of phenol observed in Kim et al.’s experiment [28,29].

Figure 5 shows the PCN formation pathways of subsequent reactions of IM28 initiated from the cross-condensation of PhR with 3-CPR. In Figure 5, all the pathways are ended with the H elimination steps. Pathways 18 and 19 have the same rate-determining step, with a potential barrier of 27.43 kcal/moL. The rate-determining step of pathway 20 has a potential barrier of 26.31 kcal/moL. Pathways 21 and 22 have the same rate-determining step, with a potential barrier of 26.94 kcal/moL. The rate-determining step of pathway 20 has a lower potential barrier than those of pathways 18, 19, 21, and 22. In addition, the H elimination step in pathway 20 has a lower potential barrier, which is less endothermic than those in pathways 18, 19, 21, and 22. Thus, pathway 20 is energetically preferred to pathways 18, 19, 21, and 22. The main product of cross-condensation of PhR with 3-CPR is 2-MCN. However, in Kim et al.’s experiment on PCN formation using 3-CP as a precursor, both 1-MCN and 2-MCN are the main MCN products using 3-CP as precursor, and they have similar formation potential [28,29]. We infer that more 1-MCN products may come from the self-condensation of 3-CPRs. Our previous theoretical study of PCN formation through the self-condensation of 3-CPRs can provide a good explanation of 1-MCN and 2-MCN formation [32]. For the 1-MCN formation from the self-condensation of 3-CPRs, the potential barriers for the rate-determining (first 6-member ring formation) step and the Cl elimination step are 22.35 and 11.20 kcal/moL, respectively [32]. For the 2-MCN formation from the self-condensation of 3-CPRs, the potential barriers for the rate-determining (first 6-member ring formation) step and the Cl elimination step are 23.17 and 12.39 kcal/moL, respectively [32]. Thus, 1-MCN formation has much more formation potential than that for 2-MCN from the self-condensation of 3-CPRs, based on our previous theoretical study [32]. In summary, 2-MCN has more formation potential than 1-MCN from the cross-condensation of PhR with 3-CPR, and 1-MCN has more formation potential than 2-MCN from the self-condensation of 3-CPRs. This can provide a reasonable explanation for the observation in the experiment that the yield of 1-MCN is almost equal to that of 2-MCN using 3-CP as precursor. Besides MCN, naphthalene was also largely observed in Kim et al.’s experiment using 3-CP as a precursor [28,29], while according to our previous theoretical study, there is no route for the formation of naphthalene from the self-condensation of 3-CPRs [32]. As shown in Figure 4, naphthalene is the main product of the subsequent reaction of IM13. This study proves that naphthalene can be produced from the cross-condensation of PhR and 3-CPR. In addition, the naphthalene formation observed in Kim et al.’s experiment may also have been caused by the self-condensation of PhRs [28,29].

#### 3.3.2. Formation of PCNs from Pursuant Reactions of Chloro-Bicyclopentadienyls by First H Shift and Then Abstraction Routes

In Figure 2, Figure 3, Figure 4 and Figure 5, the direct H abstraction steps require crossing smaller potential barriers (−1.24~5.84 kcal/moL) and strong reaction exothermics (−41.58~−25.29 kcal/moL), while in Figure 6, the H shift requires crossing a larger potential barrier (21.76~26.12 kcal/moL) and a weak reaction exothermic event (−3.26~1.22 kcal/moL). Therefore, the “direct abstraction” mechanisms shown in Figure 2, Figure 3, Figure 4 and Figure 5 are energetically favored over the “first H shift, then abstraction” mechanisms proposed by Kim et al., as shown in Figure 6 [28,29,30].

#### 3.3.3. Comparing the Cross-Condensation of PhR with 2-CPR/3-CPR and Self-Condensation of 2-CPRs/3-CPRs

A comparison of the formation of PCNs from the cross-condensation of PhR with 2-CPR/3-CPR with our previous studies of PCN formation from the self-condensation of 2-CPRs/3-CPRs [31,32] shows that the position of chlorination affects PCN formation potential. For example, the largest potential barrier to PCN formation from the cross-condensation of PhR with 2-CPR is 48.18 kcal/moL, which is higher than the value of 42.83 kcal/moL from self-condensation of 2-CPRs. This indicates that PCN formation from self-condensation of 2-CPRs is energetically preferred to that from the cross-condensation of PhR with 2-CPR. Moreover, the largest potential barrier to PCN formation from the cross-condensation of PhR with 3-CPR is 46.79 kcal/moL, which is lower than the value of 47.01 kcal/moL obtained from the self-condensation of 3-CPRs. It should be noted that the PCN formation from the cross-condensation of PhR with 3-CPR is favored over that from the self-condensation of 3-CPRs. To determine the cross-condensation of PhR with 2-CPR, it is necessary to compare it with the cross-condensation of PhR with 3-CPR. The largest potential barrier of PCN formation from the cross-condensation of PhR with 2-CPR is higher than that from the cross-condensation of PhR with 3-CPR, which means that PCN formation from the cross-condensation of PhR with 3-CPR can occur much easier than that resulting from the cross-condensation of PhR with 2-CPR.

### 3.4. Rate Constant Calculations

In this paper, the CVT/SCT rate constants are well matched to the thermodynamic analysis. For example, as presented in Appendix A, at 800 K, our CVT/SCT values for the rate-determining steps in pathways 1–4 are 9.04 × 10^−2^, 2.44 × 10^−4^, 9.55 × 10^−5^, and 6.02 × 10^−5^ s^−^^1^, which again demonstrates that pathway 1 is the most favorable for PCN formation from the cross-condensation of PhR and 2-CPR. Similarly, at 1000 K, the CVT/SCT values for the rate determining steps in pathways 5–8 are 7.53 × 10^0^, 3.27 × 10^0^, 4.25 × 10^1^, and 9.66 × 10^0^ s^−^^1^, which indicates that pathway 7 is most favorable for PCN formation from the cross-condensation of PhR and 3-CPR.

The thermodynamic analysis of PCN formation from IM5 in Figure 3 shows the rate-determining step of pathways 13–14 has a lower potential barrier than those of pathways 9–12. Comparing the calculated CVT/SCT rate constants of the rate-determining steps in these pathways also proved this conclusion. For example, in Appendix A, at 1000 K, the CVT/SCT rate constant for the rate-determining step in pathways 13–14 is 1.34 × 10^9^ (TS53) s^−1^, which is larger than the value of 5.49 × 10^8^ (TS36) s^−1^ for the rate-determining step of pathways 9–10, and 1.72 × 10^7^ s^−1^ (TS49) for pathways 11–12, respectively. Analogously, the 1000 K CVT/SCT rate constant for the rate-determining step in pathways 20 is 2.01 × 10^7^ s^−1^ (TS72), which is larger than the value of 1.05 × 10^7^ s^−1^ (TS43), and 7.84 × 10^6^ s^−1^ (TS79) for pathways 18–19 and pathways 21–22, respectively. This perfectly matches the thermodynamic analysis in Figure 5, where pathway 20 is kinetically more efficient than pathways 18, 19, 21, and 22 for PCN formation from IM28.

In order to further check the validity of our conclusion, we also compare the rate constants for the last Cl/H elimination steps. For example, at 1000 K, the CVT/SCT rate constants for the last Cl elimination step in pathway 10 is 1.77 × 10^10^ s^−1^, which is 1 order of magnitude larger than that of the H elimination step in pathway 9 (3.60 × 10^9^ s^−1^). This result reconfirms thermodynamic analysis that the pathways ended with the Cl elimination step are preferred to those ending with the H elimination step. From these good agreements, it is inferred that the same accuracy could be expected for the other crucial elementary reactions involved in the whole mechanism.

## 4. Materials and Methods

### 4.1. Density Functional Theory

A hybrid meta function MPWB1K, which yields uniformly good performance in quantum calculations for thermochemistry, thermochemical kinetics, hydrogen bonding, and weak interactions [39], was used to calculate the geometries, energies, frequencies of the reactions, products, intermediates, and transition states using the Gaussian 09 program package [40]. The geometrical parameters, harmonic vibrational frequencies, and the intrinsic reaction coordinate (IRC) were optimized at the MPWB1K level with a standard 6-31 + G(d,p) basis set. The vibrational frequencies were calculated to determine the obtained stationary points as minima, or transition states, by diagonalizing their Hessian matrices and confirming that there is no negative eigenvalue for minimas, or one negative eigenvalue for transition states, respectively. The minimum energy path (MEP) was obtained by the IRC theory to confirm that the transition state connects to minimas along the reaction path [41]. By using a more precise basis set, 6-311 + G(3df,2p), the single point energies were further carried out to obtain a more reliable evaluation of potential barriers and reaction heat. The reliability and accuracy of MPWB1K6-311 + G(3df,2p)//MPWB1K/6-31 + G(d,p) level for the geometries, vibrational frequencies, and energy calculation of PCN formation from the CPs have been confirmed in our previous works [31,32].

### 4.2. Kinetic Calculation

To obtain the rate constants of major elementary reactions involved in this study over a wide temperature range (600–1200 K), the Polyrate 9.7 program [42] is employed using the canonical variational transition-state (CVT) theory with the small curvature tunneling (SCT) contribution [35,36,37,38]. The CVT rate constant, *k*^CVT^(*T*), at a fixed temperature (*T*) that minimizes the generalized transition-state theory rate constant, *k*^GT^(*T*, *s*), with respect to the dividing surface at *s*, is expressed as
(1)$$k^{\mathrm{CVT}}(T)=\underset{s}{\min}k^{\mathrm{GT}}(T,\;s)$$

The generalized transition-state theory rate constant *k*^GT^(*T*, *s*) for *T* and a dividing surface at *s* is
(2)$$k^{\mathrm{GT}}(T,s)=\frac{\sigma k_{B}T}{h}\frac{Q^{\mathrm{GT}}(T,s)}{\Phi^{R}(T)}e^{-V_{MEP}(s)/k_{B}T}$$
where, *σ* is the symmetry factor accounting for the possibility of more than one symmetry-related reaction path, *k_B_* is Boltzmann’s constant, and *h* is Planck’s constant. Φ*^R^*(*T*) is the reactant partition function per unit volume, excluding symmetry numbers for rotation, and *Q*^GT^(*T*, *s*) is the partition function of a generalized transition state at *s*, with a local zero of energy at *V_MEP_*(*s*) and with all rotational symmetry numbers set to unity. The rotational and translation partition functions are calculated classically. Most of the vibrational modes are treated as quantum mechanical separable harmonic oscillators, whereas the low-frequency modes which correspond to hindered rotations are treated as hindered rotors.

## 5. Conclusions

(1)The formation of PCNs from the cross-condensation of PhR with 2-CPR/3-CPR contains two processes: the formation of chloro-dihydrofulvlene from the cross-condensation of PhR with 2-CPR/3-CPR, and PCN formation from subsequent reactions of chloro-bicyclopentadienyl. Pathways terminated with Cl elimination (pathways for the naphthalene formation) are preferred over those terminated with H elimination (pathways for the MCN formation). The “direct abstraction” mechanism is energetically favored over the “first H shift, then abstraction” mechanisms.(2)Naphthalene observed in Kim et al.’s experiment can be produced from the cross-condensation of PhR with 2-CPR/3-CPR. The main MCN product from the cross-condensation of PhR with 2-CPR is1-MCN, and 2-MCN is the main MCN product from the cross-condensation of PhR with 3-CPR [28,29].(3)The theoretical calculation of the PCN formation mechanism from the cross-condensation of PhR with 2-CPR/3-CPR, as well as that from the self-condensation using 2-CPRs/3-CPRs [31], can provide a reasonable explanation of the experimental observations [28,29] that the formation potential of naphthalene is larger than that of 1-MCN from 2-CP as a precursor and almost equal to 1-MCN, and 2-MCN can be produced from 3-CP as a precursor.(4)PCN formation from the self-condensation of 2-CPRs is more energetically favorable than that from the cross-condensation of PhR with 2-CPR, whereas PCN formation from the cross-condensation of PhR with 3-CPR is favored over that from the self-condensation of 3-CPRs. PCN formation from the cross-condensation of PhR with 3-CPR can occur much easier than that from the cross-condensation of PhR with 2-CPR.

## Figures and Tables

**Figure 1 ijms-23-05866-f001:**
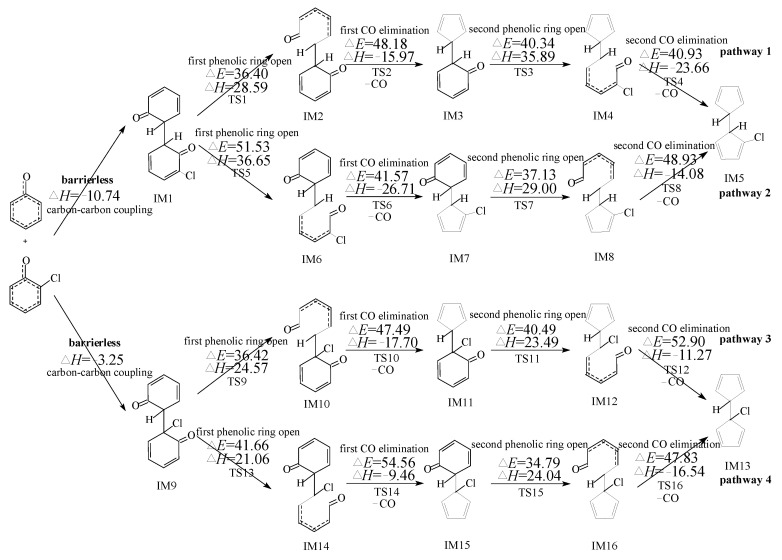
Chlorinated bicyclopentadienyl formation routes embedded with the potential barriers Δ*E* (in kcal/moL) and reaction heats Δ*H* (in kcal/moL) from the dimerization of PhR and 2−CPR. Δ*H* is calculated at 0 K.

**Figure 2 ijms-23-05866-f002:**
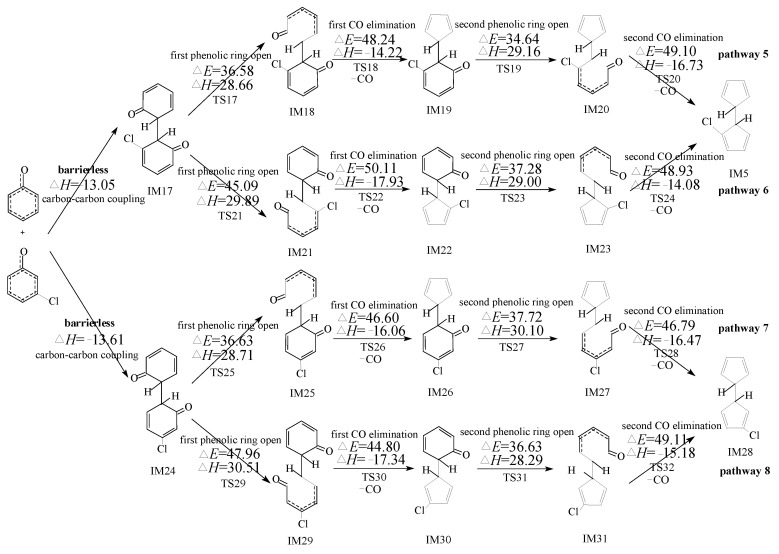
Chlorinated bicyclopentadienyl formation routes embedded with the potential barriers Δ*E* (in kcal/moL) and reaction heats Δ*H* (in kcal/moL) from the dimerization of PhR and 3−CPR. Δ*H* is calculated at 0 K.

**Figure 3 ijms-23-05866-f003:**
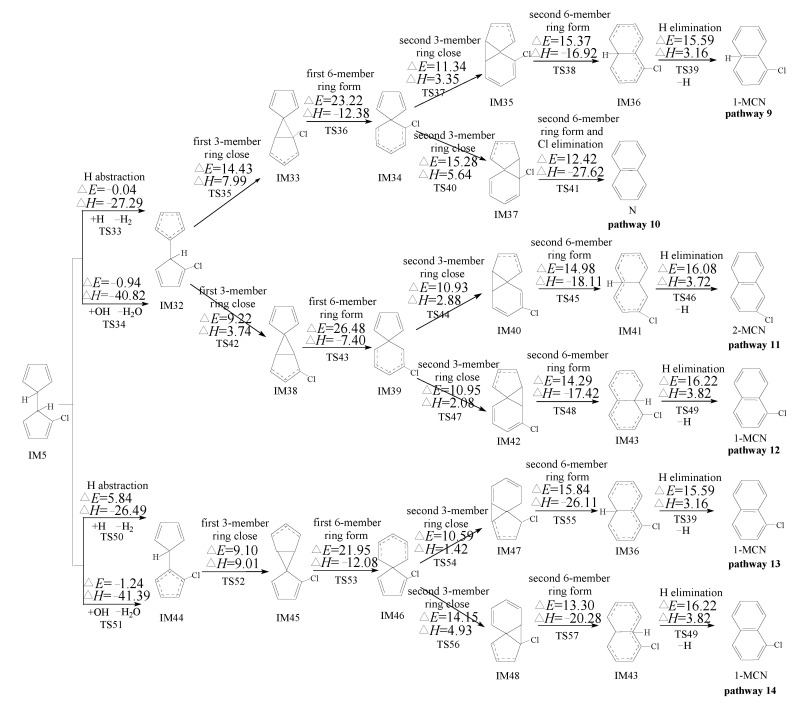
PCN formation routes embedded with the potential barriers Δ*E* (in kcal/moL) and reaction heats Δ*H* (in kcal/moL) from IM5. Δ*H* is calculated at 0 K.

**Figure 4 ijms-23-05866-f004:**
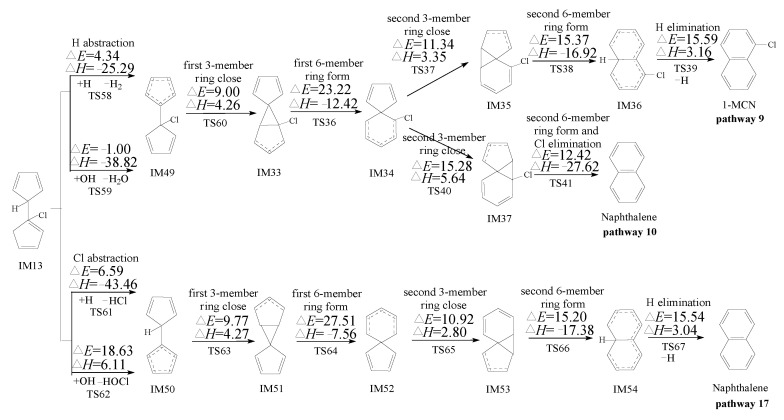
PCN formation routes embedded with the potential barriers Δ*E* and reaction heats Δ*H* from IM13.

**Figure 5 ijms-23-05866-f005:**
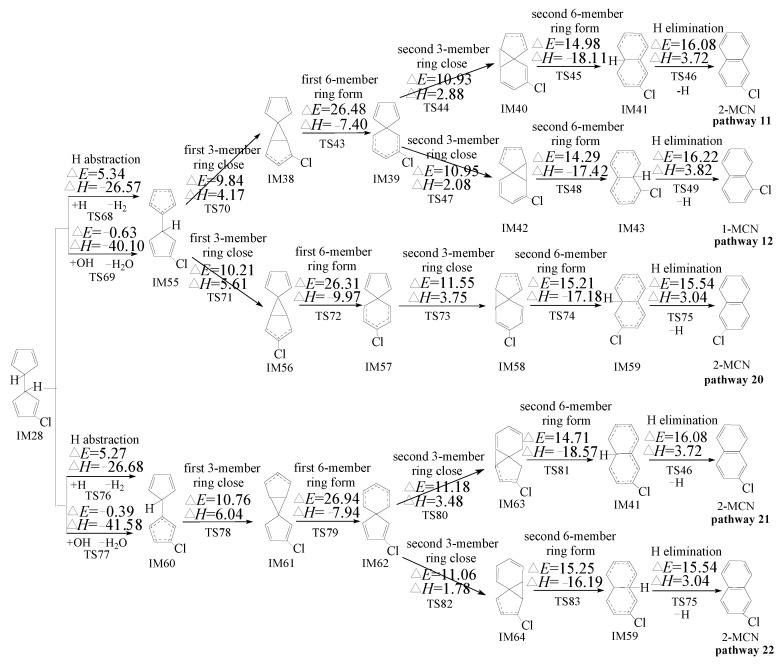
PCN formation routes embedded with the potential barriers Δ*E* and reaction heats Δ*H* from IM28.

**Figure 6 ijms-23-05866-f006:**
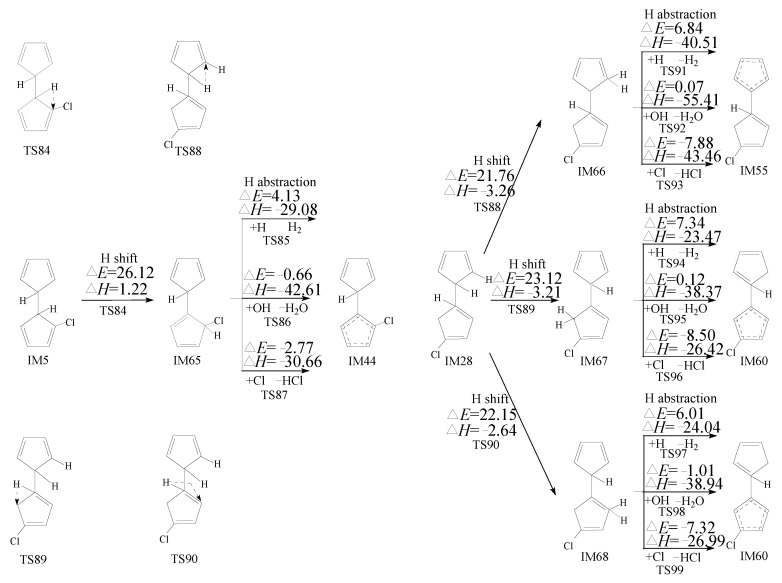
PCN formation routes embedded with the potential barriers Δ*E* (kcal/moL) and reaction heats Δ*H* (kcal/moL) from IM5 and IM28 starting with the H−shift step. Δ*H* is calculated at 0 K.

**Figure 7 ijms-23-05866-f007:**
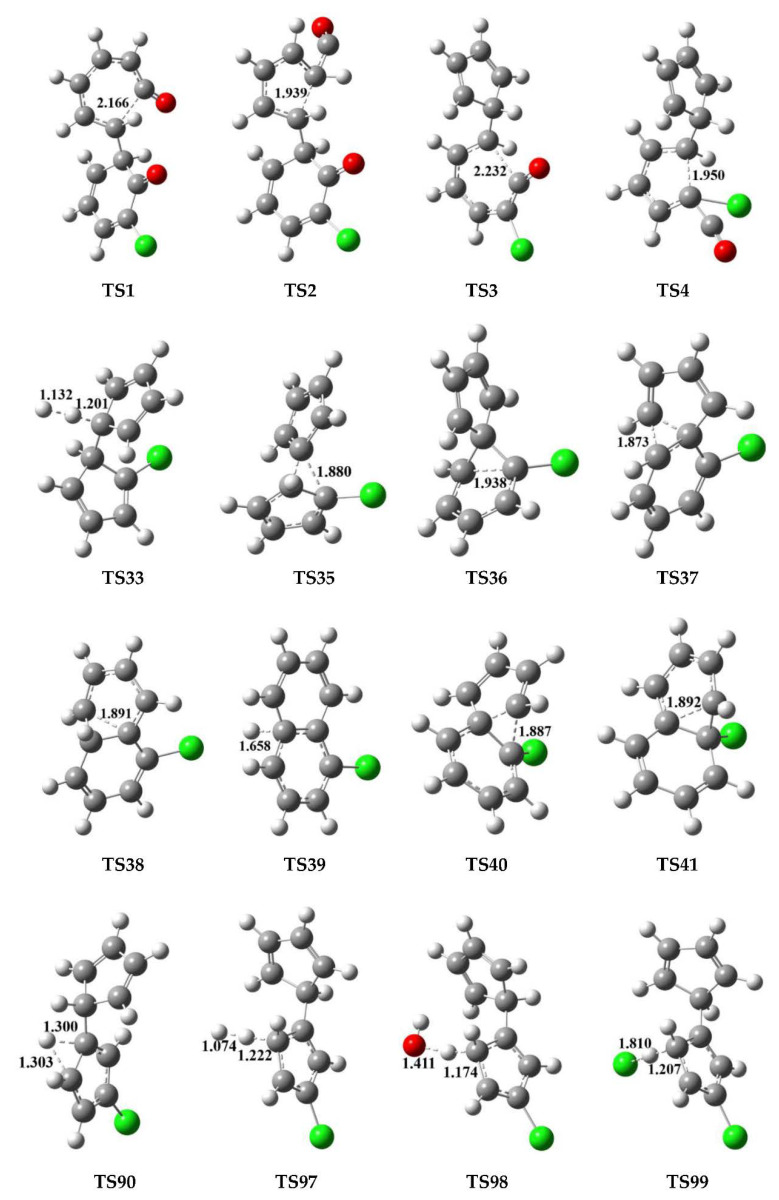
Configurations of the transition states involved in one typical route of PCN formation. Distances are in angstroms. Gray sphere, C; White sphere, H; Red sphere, O; Green sphere, Cl. (For interpretation of the references to color in this figure legend, the reader is referred to the web version of this article).

**Table 1 ijms-23-05866-t001:** Arrhenius formulas for chloro-bicyclopentadienyls formation routes from the cross-condensation reactions of PhR with 2-CPR and 3-CPR over the temperature range of 600~1200 K. (Units are s^−1^ and cm^3^ molecule^−1^ s^−1^ for unimolecular and bimolecular reactions, respectively).

Reactions Arrhenius Formulas	Arrhenius Formulas
IM1→IM2 via TS1	*k*(T) = (2.54 × 10^13^) exp(−19,455.46/T)
IM2→IM3 + CO via TS2	*k*(T) = (8.29 × 10^10^) exp(−22,071.16/T)
IM3→IM4 via TS3	*k*(T) = (3.05 × 10^13^) exp(−21,165.47/T)
IM4→IM5 + CO via TS4	*k*(T) = (6.50 × 10^11^) exp(−20,913.80/T)
IM1→IM6 via TS5	*k*(T) = (2.72 × 10^11^) exp(−27,739.54/T)
IM6→IM7 + CO via TS6	*k*(T) = (1.59 × 10^12^) exp(−20,905.12/T)
IM7→IM8 via TS7	*k*(T) = (2.08 × 10^13^) exp(−19,698.59/T)
IM8→IM5 + CO via TS8	*k*(T) = (2.20 × 10^11^) exp(−24,734.37/T)
IM9→IM10 via TS9	*k*(T) = (1.78 × 10^13^) exp(−19,076.64/T)
IM10→IM11 + CO via TS10	*k*(T) = (3.82 × 10^11^) exp(−24,130.47/T)
IM11→IM12 via TS11	*k*(T) = (2.06 × 10^13^) exp(−21,131.16/T)
IM12→IM13 + CO via TS12	*k*(T) = (3.29 × 10^11^) exp(−28,711.62/T)
IM9→IM14 via TS13	*k*(T) = (2.34 × 10^13^) exp(−21,481.07/T)
IM14→IM15 + CO via TS14	*k*(T) = (2.14 × 10^11^) exp(−28,645.35/T)
IM15→IM16 via TS15	*k*(T) = (1.54 × 10^13^) exp(−18,260.72/T)
IM16→IM13 + CO via TS16	*k*(T) = (4.97 × 10^11^) exp(−24,263.04/T)
IM17→IM18 via TS17	*k*(T) = (2.40 × 10^13^) exp(−19,312.41/T)
IM18→IM19 + CO via TS18	*k*(T) = (4.42 × 10^11^) exp(−24,236.82/T)
IM19→IM20 via TS19	*k*(T) = (2.59 × 10^13^) exp(−18,506.41/T)
IM20→IM5 + CO via TS20	*k*(T) = (6.45 × 10^11^) exp(−25,174.58/T)
IM17→IM21 via TS21	*k*(T) = (4.05 × 10^13^) exp(−23,466.51/T)
IM21→IM22 + CO via TS22	*k*(T) = (4.22 × 10^11^) exp(−25,584.87/T)
IM22→IM23 via TS23	*k*(T) = (1.00 × 10^11^) exp(−4309.34/T)
IM23→IM5 + CO via TS24	*k*(T) = (2.23 × 10^11^) exp(−24,680.07/T)
IM24→IM25 via TS25	*k*(T) = (1.52 × 10^12^) exp(−19,589.40/T)
IM25→IM26 + CO via TS26	*k*(T) = (2.15 × 10^11^) exp(−23,682.22/T)
IM26→IM27 via TS27	*k*(T) = (1.42 × 10^13^) exp(−19,927.13/T)
IM27→IM28 + CO via TS28	*k*(T) = (9.55 × 10^11^) exp(−23,833.30/T)
IM24→IM29 via TS29	*k*(T) = (2.69 × 10^13^) exp(−25,090.63/T)
IM29→IM30 + CO via TS30	*k*(T) = (1.40 × 10^10^) exp(−21,048.41/T)
IM30→IM31 via TS31	*k*(T) = (7.50 × 10^12^) exp(−19,656.05/T)
IM31→IM28 + CO via TS32	*k*(T) = (6.05 × 10^11^) exp(−24,859.70/T)

**Table 2 ijms-23-05866-t002:** Arrhenius formulas for chloro-bicyclopentadienyls formation routes from the cross-condensation reactions of PhR with 2-CPR over the temperature range of 600~1200 K. (Units are s^−1^ and cm^3^ molecule^−^^1^ s^−^^1^ for unimolecular and bimolecular reactions, respectively).

Reactions Arrhenius Formulas	Arrhenius Formulas
IM5 + H→IM32 + H_2_ via TS33	*k*(T) = (9.91 × 10^−12^) exp(−1436.77/T)
IM32→IM33 via TS35	*k*(T) = (2.30 × 10^12^) exp(−18,463.38/T)
IM33→IM34 via TS36	*k*(T) = (5.29× 10^12^) exp(−9166.55/T)
IM34→IM35 via TS37	*k*(T) = (5.87× 10^12^) exp(−6278.72/T)
IM35→IM36 via TS38	*k*(T) = (4.51 × 10^12^) exp(−8007.32/T)
IM36→1–MCN + H via TS39	*k*(T) = (1.80 × 10^13^) exp(−8511.99/T)
IM34→IM37 via TS40	*k*(T) = (1.48 × 10^12^) exp(−8686.54/T)
IM37→naphthalene via TS41	*k*(T) = (1.96 × 10^13^) exp(−7007.83/T)
IM32→IM38 via TS42	*k*(T) = (7.48 × 10^11^) exp(−7882.77/T)
IM38→IM39 via TS43	*k*(T) = (1.80 × 10^13^) exp(−14,352.16/T)
IM39→IM40 via TS44	*k*(T) = (1.94 × 10^12^) exp(−5928.21/T)
IM40→IM41 via TS45	*k*(T) = (5.17 × 10^12^) exp(−7830.77/T)
IM41→2-MCN + H via TS46	*k*(T) = (7.07 × 10^12^) exp(−8307.29/T)
IM39→IM42 via TS47	*k*(T) = (2.37 × 10^12^) exp(−5684.42/T)
IM42→IM43 via TS48	*k*(T) = (7.61 × 10^12^) exp(−7631.97/T)
IM43→1-MCN + H via TS49	*k*(T) = (1.72 × 10^13^) exp(−8895.75/T)
IM5 + H→IM44 + H_2_ via TS50	*k*(T) = (6.74 × 10^−11^) exp(−3328.55/T)
IM44→IM45 via TS52	*k*(T) = (2.49 × 10^12^) exp(−5056.83/T)
IM45→IM46 via TS53	*k*(T) = (9.36 × 10^12^) exp(−8844.00/T)
IM46→IM47 via TS54	*k*(T) = (1.74 × 10^12^) exp(−5737.01/T)
IM47→IM36 via TS55	*k*(T) = (6.33 × 10^12^) exp(−5056.83/T)
IM46→IM48 via TS56	*k*(T) = (9.72 × 10^11^) exp(−7692.64/T)
IM48→IM43 via TS57	*k*(T) = (1.12 × 10^13^) exp(−7110.50/T)
IM13 + H→IM49 + H_2_ via TS58	*k*(T) = (1.87 × 10^−11^) exp(−2331.16/T)
IM49→IM33 via TS60	*k*(T) = (4.19 × 10^11^) exp(−5100.98/T)
IM13 + H→IM50 + HCl via TS61	*k*(T) = (6.66 × 10^−11^) exp(−4051.86/T)
IM13 + OH→IM50 + HOCl via TS62	*k*(T) = (2.90 × 10^−11^) exp(−6850.19/T)
IM50→IM51 via TS63	*k*(T) = (8.57 × 10^11^) exp(−5187.97/T)
IM51→IM52 via TS64	*k*(T) = (1.60 × 10^12^) exp(−14,041.51/T)
IM52→IM53 via TS65	*k*(T) = (1.14 × 10^12^) exp(−6156.97/T)
IM53→IM54 via TS66	*k*(T) = (4.70 × 10^12^) exp(−7819.81/T)
IM54→naphthalene + H via TS67	*k*(T) = (1.61 × 10^13^) exp(−8459.84/T)
IM28 + H→IM55 + H_2_ via TS68	*k*(T) = (5.16 × 10^−11^) exp(−3030.02/T)
IM28 + OH→IM55 + H_2_O via TS69	*k*(T) = (9.71 × 10^−14^) exp(−2135.19/T)
IM55→IM38 via TS70	*k*(T) = (1.36 × 10^12^) exp(−5053.71/T)
IM55→IM56 via TS71	*k*(T) = (1.21 × 10^12^) exp(−5103.87/T)
IM56→IM57 via TS72	*k*(T) = (1.90 × 10^13^) exp(−13,757.06/T)
IM57→IM58 via TS73	*k*(T) = (2.88 × 10^12^) exp(−8011.41/T)
IM58→IM59 via TS74	*k*(T) = (1.86 × 10^13^) exp(−8105.33/T)
IM59→2-MCN + H via TS75	*k*(T) = (1.13 × 10^13^) exp(−4381.07/T)
IM28 + H→IM60 + H_2_ via TS76	*k*(T) = (2.77 × 10^−12^) exp(−4600.55/T)
IM60→IM61 via TS78	*k*(T) = (8.07 × 10^11^) exp(−5830.65/T)
IM61→IM62 via TS79	*k*(T) = (1.00 × 10^13^) exp(−14,059.47/T)
IM62→IM63 via TS80	*k*(T) = (2.01 × 10^12^) exp(−6103.57/T)
IM63→IM41 via TS81	*k*(T) = (7.30 × 10^12^) exp(−7837.39/T)
IM62→IM64 via TS82	*k*(T) = (1.99 × 10^12^) exp(−5912.17/T)
IM64→IM59 via TS83	*k*(T) = (1.50 × 10^13^) exp(−8098.93/T)

## Data Availability

The data that support the findings of this study are available from the corresponding author upon reasonable request.

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
