# Peer review of "The Homogeneous Gas-Phase Formation Mechanism of PCNs from Cross-Condensation of Phenoxy Radical with 2-CPR and 3-CPR: A Theoretical Mechanistic and Kinetic Study"

_ijms, 2022, doi:10.3390/ijms23115866_

Round 1
Reviewer 1 Report
The main aim of this study is to carry out detailed thermodynamics and kinetic calculations in order to investigate the PCN (polychlorinated naphthalene) formation mechanisms from PhR (phenoxy radical) with 2-CPR/3-CPR (chlorophenoxy radical). PCN are the products obtained upon naphthalene treatment with chlorine. It is a toxic chemical compound, hazardous to humans. While it has been used widely in insulating coatings for electrical wires in the past, currently it is being phased out. However, it may also be created naturally during wildfires and other mechanisms. For these reasons it is important to study more extensively the PCN formation mechanisms in order to further our understanding of this toxic product. This study has done exactly that, focusing the main attention to the formation mechanisms including PhR and CPR. It is a very thorough theoretical mechanistic and kinetic study, providing the reader with a lot of information regarding formation pathways and mechanisms, backed up by theoretical calculations. However, it is wort mentioning that there are a few continuity errors in the article, along with some spelling mistakes. In the discussion of the results the text rapidly becomes cumbersome and difficult to understand. A lot of raw technical data is making it hard to follow the steps of the study.
The mechanisms were compared with the Kim’s experimental study. The results can give some explanations of the experimental observations by analyzing the formation potentials. The study has a lot of computations but there are some issues which must be explained.
Some remarks:
- There is a need for more details how the computations where performed? What does it mean “H shift”? From real word perspective how the H shift should be formed? Is they realistic? The figures 1-6 must be improved.
- The Kims‘ experiment was done in 2005. Why it is important? The short summary about it should be done as there the research has a background on it.
- How was generated one hundred TS (TS1-TS99) structures? Is it a mistake? In Table 1-3 as well as in supplementary there is lack of TS84-TS99 structures. It is unclear how the table was generated. Was there used any background from other literature? Was it somehow generated?
- Line 16 – “… are abundant in thermal and combustion procedures…” , what kind of thermal and combustion procedures? Abundant in all combustion procedures no matter the reductant? It is later explained in the introduction, but not in the abstract;
- What is the main reason for this study? Polychlorinated naphthalene (PCN) is a hazardous material, what is the main aim of this study regarding PCN?
- Line 26 – what is N and 1-MCN/ 2-MCN, monochlorinated naphthalene? It is later explained in the introduction, but not in the abstract;
- Line 33 – “…and almost equal 1-MCN and 2-MCN can be produced…”, equal what? Equal quantities, amounts, yelds?
- Line 96 – “However, despite lots of experimental observations can be...” – questionable usage of ‘Despite’, Could be “Despite the fact that lots..”;
- Line 123 – questionable continuity, PhR is called phenoxy radical, yet CPR is called chlorophenoxy. Shouldn’t the word ‘radical’ be used as well?
- Line 138/139– “dihydrofulvlenes” probable spelling mistake;
- Line 303 – “…can occur intramolecular rearrangement...” , possibly incorrect usage of “occur”, possible changes “cause”, “induce”, “leads to”;
- Line 311 – “…has higher potential barrier and less exothermic than…”, possibly missing verb “is less exothermic”;
- Line 358 – “Hessian matrixes”, à matrices;
- Line 412 – “In Figure 2, there exit no…” , exists? “Exit” is either a wrong word, or used incorrectly;
- Lines 504-509 – probably incorrect usage of “exothermic” and “endothermic”, lacks additional supporting words like “reaction”;
Author Response
Thank you very much for your comments. A point-by-point response to your comments is upload as a word file. Please see the attachment.

Reviewer 2 Report
The paper “The Homogeneous Gas-phase Formation Mechanism of PCNs 2 from Cross-Condensation of phenoxy radical with 2-CPR and 3- 3 CPR: A Theoretical Mechanistic and Kinetic Study” by Zhuochao Teng, Yanan Han, Shuming He, Xianwei Zhao, Qi Zhang, Xurong Bai, Xiaotong Wang, Yanhui Sun, and Fei Xu, uses computational theoretical methods to perform a detailed study of the reaction pathways of polychlorinated naphthalenes formation from cross-condensation of phenoxy radical with chlorophenoxy radicals. The paper is mostly well written and it presents a substantial amount of data. Throughout the paper the author shows the correlation of their theoretical results with experimental ones published elsewhere. The only worry is about the paper the statements in
Page 3 line 133 the authors mention that “the potential barriers of 133 CPs and phenol with OH/Cl radicals are negative due to the ZPE…” and Page 5 line 226 “abstraction step of chloro-dihydrofulvenes by OH are negative due to 226 the ZPE (zero-point energy) correction and existence of pre-reactive complexes.” they are very strong and non-trivial affirmation and I believe that it requires a better justification and references. If the authors adequately address these points the paper deserves to be published undoubtedly.
Below are itemized few minor concerns in the paper structure that can be easily fixed by the authors.
- In page 3 line 99 the authors introduce the acronym N to refer to naphthalene, however the N letter is the standard symbol of the Nitrogen atom, in order to avoid misconceptions it would be better replace the acronym used to describe the naphthalene molecule in the text.
- Page 3 line 133 the authors mention that “the potential barriers of 133 CPs and phenol with OH/Cl radicals are negative due to the ZPE…” The \Delta symbol in all figures in the paper must be enlarged for better quality presentation.
Author Response

(The authors gave the same response as above.)

Round 2
Reviewer 1 Report
The authors addressed and explained in details all questions. I think it can be published as it is.